# MDP Playground: A Design and Debug Testbed for Reinforcement Learning

**Raghu Rajan**[1], **Jessica Lizeth Borja Diaz**[1], **Suresh Guttikonda**[1],
**Fabio Ferreira**[1], **André Biedenkapp**[1], **Jan Ole von Hartz**[1] **& Frank Hutter**[1,2]
[1] University of Freiburg [2] Bosch Center for Artificial Intelligence
rajanr@cs.uni-freiburg.de

## Abstract

We present *MDP Playground*, an efficient testbed for Reinforcement Learning
(RL) agents with *orthogonal* dimensions that can be controlled independently
to challenge agents in different ways and obtain varying degrees of hardness in
generated environments. We consider and allow control over a wide variety of
dimensions, including *delayed rewards*, *rewardable sequences*, *density of rewards*,
*stochasticity*, *image representations*, *irrelevant features*, *time unit*, *action range*
and more. We define a parameterised collection of fast-to-run toy environments
in *OpenAI Gym* by varying these dimensions and propose to use these for the
initial design and development of agents. We also provide wrappers that inject
these dimensions into complex environments from *Atari* and *Mujoco* to allow for
evaluating agent robustness. We further provide various example use-cases and
instructions on how to use *MDP Playground* to design and debug agents. We
believe that *MDP Playground* is a valuable testbed for researchers designing new,
adaptive and intelligent RL agents and those wanting to unit test their agents.

## 1 Introduction

RL has succeeded at many disparate tasks, such as helicopter aerobatics, game-playing and continuous
control [2, 38, 49, 10, 14, 17]. However, a lot of the insights obtained are on very complex and in
many instances *blackbox* environments.

There are many different types of standard environments, as many as there are different kinds of
tasks in RL [e.g. 57, 6, 11]. They specialise in *specific* kinds of tasks. The underlying assumptions
in many of these environments are that of a Markov Decision Process (MDP) [see, e.g., 44, 52] or
a Partially Observable MDP (POMDP) [see, e.g., 22, 25]. However, there is a lack of simple and
*general* MDPs which capture common difficulties seen in RL and let researchers experiment with
them in a fine-grained manner. Many researchers design their own toy problems which capture the
key aspect of their problem and then try to gain *whitebox* insights because the standard complex
environments, such as *Atari* and *Mujoco*, are too expensive or too opaque for the initial design and
development of their agent. To standardise this initial design and debug phase of the development
pipeline, we propose a platform which *distils* difficulties for MDPs that can be generalised across RL
problems and allows to *independently* inject these difficulties.

Disadvantages of *complex* environments when considered from a point of view of a design and
debug testbed include: **1)** They are very expensive to evaluate. For example, a DQN [38] run on
*Atari* [6] took us 4 CPU days and 64GB of memory to run. **2)** The environment structure itself is
so complex that it leads to "lucky" agents performing better (e.g., in [18]). Furthermore, different
implementations even using the same libraries can lead to very different results [18]. **3)** Many
difficulties are concurrently present in the environments and do not allow us to independently test

Submitted to the 35th Conference on Neural Information Processing Systems (NeurIPS 2021) Track on Datasets
and Benchmarks. Do not distribute.

their impact on agents' performance. During the design phase, we need environments to encapsulate, preferably orthogonally, the different difficulties present. For instance, MNIST [32] captured some key difficulties required for computer vision (CV) which made it a good testbed for designing and debugging CV algorithms, even though it cannot be used to directly learn models for much more specific CV applications such as classification of plants or medical image analysis.

The main contributions of this paper are:

- We identify and discuss dimensions of MDPs that can have a significant effect on agent performance, both for discrete and continuous environments;
- We discuss how to use *MDP Playground* to design and debug agents with various experiments; toy experiments can be run in as few as 30 seconds on a single core of a laptop;
- We discuss insights that can be gained with the various considered dimensions; transferring insights from toy to complex environments for some under-studied dimensions led to significant improvements in performances on complex environments.

## 2 Dimensions of MDPs

We try to exhaustively identify orthogonal *dimensions* of hardness in RL by going over the many components of a (PO)MDP. By *orthogonal*, we mean that these dimensions are present independent of each other in environments. This was tried *exhaustively* to allow as many dimensions as possible for researchers to systematically study them and gain new insights.

We define an MDP as a 7-tuple $(S, A, P, R, \rho_o, \gamma, T)$, where $S$ is the set of states, $A$ is the set of actions, $P : S \times A \rightarrow S$ describes the transition dynamics, $R : S \times A \times S \rightarrow \mathbb{R}$ describes the reward dynamics, $\rho_o : S \rightarrow \mathbb{R}^+$ is the initial state distribution, $\gamma$ is the discount factor and $T$ is the set of terminal states. We define a POMDP with two additional components - $O$ represents the set of observations and $\Omega : S \times A \times O \rightarrow \mathbb{R}^+$ describes the probability density function of an observation given a state and action. To clarify terminology, following [51] we will use *information state* to mean the state representation used by the agent and *belief state* as the posterior belief of the unobserved state given the full observation history. If the belief state were to be used as the information state by an agent, this would be sufficient to compute an optimal policy. However, since the full observation history is not tractable to store for many environments, agents in practice use the last few observations as their information state which renders it only partially observable. This is important because many of the motivated dimensions are actually due to the information state being non-Markov.

### 2.1 MDPs in MDP Playground

**Toy Environments** The toy environments are cheap and encapsulate all the identified dimensions. The components of the MDP can be automatically generated according to the dimensions or can be user-defined. Any dimension not specified is set to a vanilla default value. Further, the underlying MDP state is exposed in an *augmented_state* variable, which allows users to design agents that may try to identify the true underlying MDP state given the observations. We now briefly describe the auto-generated discrete and continuous environments, since we use these for the experiments section and expect that these will cover the majority of the use-cases. This is followed by implementation details of selected dimensions; details for all dimensions can be found in Algorithm 1 in Appendix C.

**Discrete Environments** In the discrete case, $S$ and $A$ contain *categorical* elements, and random instantiations of $P$ and $R$ are generated after the remaining dimensions have been set. The generated $P$ and $R$ are deterministic and held fixed for the environment. We keep $\rho_o$ to be uniform over the non-terminal states, and $T$ is fixed to be a subset of $S$ based on a chosen *terminal state density*.

**Continuous Environments** In the continuous case, environments correspond to the simplest real world task we could find: moving a rigid body to a target point, similar to [16] and [28]. $P$ is formulated such that each action dimension affects the corresponding space dimension - $s$ is set to be equal to the action applied for *time unit* seconds on a rigid body. This is integrated over time to yield the next state. $R$ is designed such that the reward for the current time step is the distance travelled towards the target since the last step.

Both, the discrete and continuous environments, in *MDP Playground* can be described as graphical POMDPs.

## 2.2 Motivations of Dimensions and Implementations

We now describe many of the dimensions from a general point of view and their implementations in *MDP Playground*. For clarity, we describe only the dimensions with experiments in the main paper here in greater detail and refer the reader to Appendix B and the documentation for more detailed descriptions of all the dimensions.

**Reward Delay** For many environments, in many situations, agents perform an action that is consequential to receiving a reward but the agent is only rewarded in a *delayed* manner [see e.g. 4] (see Figure 1d). For example, shooting at an enemy ship in *Space Invaders* leads to rewards much later than the action of shooting. Any action taken after that is inconsequential to obtaining the reward for destroying that enemy ship. In *MDP Playground*, the reward is artificially delayed by a non-negative integer number of timesteps, $d$.

**Reward Density** Environments can also be characterised by their *reward density*. When an environment has denser rewards (see Figure 1a), one is more likely to obtain a supervisory reward signal. In sparse reward settings [15], the reward is 0 more frequently, especially, for example, in continuous control environments where a long trajectory is followed and then a single non-zero reward is received at its end. In *MDP Playground*, for discrete environments, the *reward density*, $rd$, is defined as the fraction of possible sequences of length $n$ that are actually rewarded by the environment, given that $n$ is constant. If $num_r$ sequences are rewarded, we define the reward density to be $rd = num_r / \frac{(|S|-|T|)!}{(|S|-|T|-n)!}$ and the sparsity as $1 - rd$. For continuous environments, density is controlled by having a sparse or dense environment using a *make_denser* configuration option.

**Stochasticity** Another characteristic of environments that can significantly impact performance of agents is *stochasticity*. The environment, i.e., dynamics $P$ and $R$, may be stochastic or may seem stochastic to the agent due to partial observability or sensor noise (see Figure 1b-1c). A robot equipped with a rangefinder, for example, has to deal with various sources of noise in its sensors [55]. In *MDP Playground*, for discrete environments, *transition noise* $t\_n \in [0, 1]$; with probability $t\_n$, an environment transitions uniformly at random to a state that is not the *true* next state given by $P$. For discrete environments, *reward noise* $r\_n \in \mathbb{R}$; a normal random variable distributed according to $\mathcal{N}(0, \sigma^2_{r\_n})$ is added to the *true* reward. For continuous environments, both $p\_n$ and $r\_n$ are normally distributed and directly added to the states and rewards.

**Irrelevant Features** Environments also tend to have a lot of *irrelevant features* [45] that one need not focus on. This holds for both table-based learners and approximators like Neural Networks (NNs). NNs additionally can even fit random noise [64] and having irrelevant features is likely to degrade performance. For example, in certain racing car games, though the whole screen is visible, concentrating on only the road would be more efficient without loss in performance. In *MDP Playground*, for discrete environments, a new discrete dimension with its own transition function $P_{irr}$ which is independent of $P$, is introduced. However, only the discrete dimension corresponding to $P$ is *relevant* to calculate the reward function. Similarly, in continuous environments, dimensions of $S$ and $A$ are labelled as irrelevant and not considered in the reward calculation.

**Representations** Another aspect is that of *representations*. The same underlying state may have many different external representations/observations, e.g., *feature* space vs *pixel* space. Mujoco tasks may be learnt in feature space vs directly from pixels, and Atari games can use the underlying RAM state or images. For images, various image transformations [*shift*, *scale*, *rotate*, *flip* and others; 19] may manifest as observations of the same underlying state and can pose a challenge to learning. In *MDP Playground*, for discrete environments, when this aspect is enabled, each categorical state is associated with an image of a regular polygon which becomes the externally visible observation $o$ to the agent. This image can further be transformed by *shifting*, *scaling*, *rotating* or *flipping*, which are applied at random to the polygon whenever an observation is generated. For continuous environments, image observations can be rendered for 2D environments. Examples of some generated states can be seen in Figures 10-11 in Appendix I.

**Time Unit and Action Range** For continuous control problems, we describe 2 additional dimensions here: *action range* [26], a weight penalising actions; and *time unit*, the discretisation of time (see Figure 1e).

We now summarise the dimensions identified above (with the (PO)MDP component they impact in brackets):

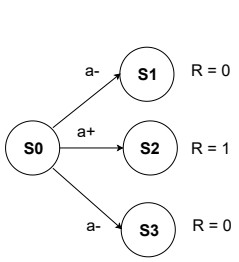

(a) $R$ density: only 1 of 3 possible actions (a+) leads to a reward

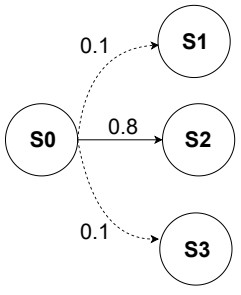

(b) $P$ noise: A noise of 0.2 (split into 0.1 and 0.1 and shown with dotted lines) is shown to lead the agent to a state which is not the true next state.

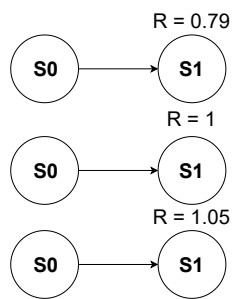

(c) $R$ noise: The *same* transition leads to different rewards.

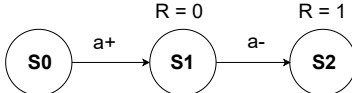

(d) $R$ delay: The rewarding action (a+) leads to a reward not immediately but a step later than it was executed and this reward is achieved even though an action inconsequential to achieving the reward (a-) was performed. Note: the reward would have been achieved a step later irrespective of which action was performed in the second step.

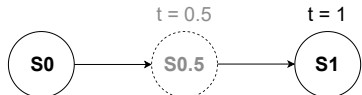

(e) Time Unit: We depict a "half" action, i.e., performed for a *time unit* that is half the default time unit, leading to an intermediate state

Figure 1: We depict some of the dimensions visually following [59]. Not all states and actions are depicted to focus on the dimension of interest. Rewarding actions are shown as a+ while actions shown as a- are not rewarding. Reward is shown as R and time unit as t.

- Reward Delay ($R$)
- Reward Density ($R$)
- Transition Noise ($P$)
- Reward Noise ($R$)
- Irrelevant Features ($O$)
- Representations ($O$)
- Action Range ($A$)
- Time Unit ($P$)

Only selected dimensions are included here, to aid in understanding and to show use-cases for *MDP playground*. Trying to exhaustively identify dimensions has led to a very flexible platform and Appendix B lists all the dimensions of MDP Playground. We would like to point out that it largely depends on the domain which dimensions are important. For instance, in a video game domain, a practitioner may not want to inject any kind of noise into the environment, if their only aim is to obtain high scores, whereas in a domain like robotics adding such noise to a deterministic simulator could be crucial in order to obtain generalisable policies [56].

## 3 MDP Playground

**Code samples** An environment instance is created as easily as passing a Python `dict`:

```
from mdp_playground.envs import RLToyEnv
config = {
    'state_space_type': 'discrete',
    'action_space_size': 8,
    'delay': 1,
    'sequence_length': 3,
    'reward_density': 0.25,
    }
env = RLToyEnv(**config)
```

**Very low-cost execution** Experiments with *MDP Playground* are cheap, allowing academics without special hardware to perform insightful experiments. Wall-clock times depend a lot on the agent, network size (in case of NNs) and the dimensions used. Nevertheless, to give the reader an idea of the runtimes involved, DQN experiments (with a network with 2 hidden layers of 256 units each) took on average 35s for a *complete* run of DQN

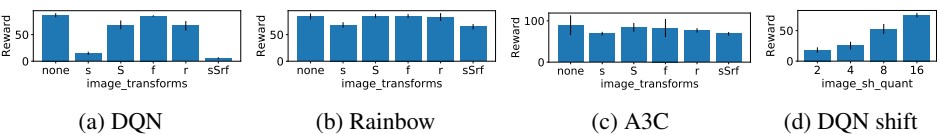

| (a) DQN | (b) Rainbow | (c) A3C | (d) DQN shift |

Figure 2: AUC of episodic reward at the end of training for the different agents **when varying representation**. 's' denotes *shift* (quantisation of 1), 'S' *scale*, 'f' *flip* and 'r' *rotate* in the labels in the first three subfigures and *image_sh_quant* represents quantisation of the *shift*s in the DQN experiment for this. Error bars represent 1 standard deviation. Note the different reward scales.

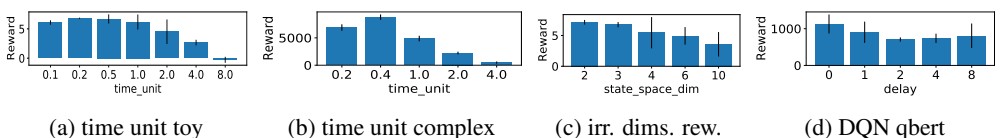

| (a) time unit toy | (b) time unit complex | (c) irr. dims. rew. | (d) DQN qbert |

Figure 3: **a** and **b**: DDPG with **time unit** on toy and complex (HalfCheetah) environment at the end of training (*time unit* is relative to the defaults). **c**: DDPG with **irrelevant dimensions** injected on the toy environment. **d**: DQN on qbert. Error bars represent 1 standard deviation. Note the different y-axis scales.

for 20 000 environment steps. In this setting, we restricted Ray RLLib [33] and the underlying Tensorflow [1] to run on *one core of a laptop* (core-i7-8850H CPU – the full CPU specifications for a single core can be found in Appendix R). This equates to roughly 30 minutes for the *entire* delay experiment shown in Figure 12a which was plotted using 50 runs (10 seeds × 5 settings for *delay*; these 50 runs could also be run in an embarrassingly parallel manner on a cluster). Even when using the more expensive continuous or representation learning environments, runs were only about 3-5 times slower.

**Complex Environment Wrappers** We further provide wrappers for *Atari* and *Mujoco* which can be used to inject some of the dimensions also into complex environments.

**Design decisions** While many dimensions can seem challenging at first, it is also the nature of RL that different dimensions tend to be important in different specific applications. The video game domain was provided as an example of this in Section 2.2. Another example is of *reward scale*. The agents we tested here re-scale or clip rewards already and the effects of this dimension are not as important as they would be otherwise. To maintain the flexibility of having as many dimensions as possible and yet keep the platform easy to use, **default** values are set for dimensions that are not configured. This effectively turns off those dimensions. Thus, as in the code example, users only need to provide dimensions they are interested in.

Further design decisions are discussed in detail in Appendix G.

# 4 Using MDP Playground

We discuss in detail various experiments along with how they may be used to design new agents and to debug existing agents. For the experiments, we set $|S|$ and $|A|$ to 8 and the *terminal state density* to 0.25. The *reward scale* is set to 1.0 whenever a reward is given by the environment. We evaluated *Rllib* implementations [33] of DQN [38], Rainbow DQN [20], A3C [37] on discrete environments and DDPG [34], TD3 [14] and SAC [17] on continuous environments over grids of values for the dimensions. Hyperparameters and the tuning procedure used are available in Appendix O. We used fully connected networks except for pixel-based representations where we used Convolutional Neural Networks (CNNs) [31].

## 4.1 Designing New Agents

We hope our toy environments will help identify inductive biases needed for designing new RL agents without getting confounded by other sources of "noise" in the evaluation. What is important for doing

this is to be able to identify if the trends seen on the toy environments would also occur for more complex environments. We now provide empirical support for this with several experiments.

We tested the trends of the dimensions on more complex Atari and Mujoco tasks. For Atari, we ran the agents on *beam_rider*, *breakout*, *qbert* and *space_invaders* when varying the dimensions *delay* and *transition noise*. For Mujoco, we ran the agents on *HalfCheetah*, *Pusher* and *Reacher* using *mujoco-py* when varying the dimensions *time unit* and *action range*. We evaluated 5 seeds for $500k$ steps for *Pusher* and *Reacher*, $3M$ for *HalfCheetah* and $10M$ ($40M$ frames) for Atari. The values shown for *action range* and *time unit* are relative to the ones used in Mujoco.

**Varying *representations*** We turned on image representations for discrete environments and applied various transforms (*shift*, *scale*, *rotate* and *flip*) one at a time and also all at once. We observed that the more transforms are applied to the images, the harder it is for agents to learn, as can be seen in Figures 2a-c. This was to be expected since there are many more combinations to generalise over for the agent.

It is important to note, from the point of view of a design platform, that our platform allows us to identify the inductive bias of CNNs being good for image observations without having to conduct such experiments on complex and expensive environments. This is because the toy environments capture many key features of image representations and thus the image classification capabilities of CNNs can help identify the underlying MDP state. In a similar manner, we have captured key features of other dimensions. If one were to design a new inductive bias which helps the agent identify the underlying MDP state in the presence of the other dimensions, this could be tested in a coarse and quick manner on our platform.

**Varying *time unit*** We observed that the *time unit* has an optimal value which has significant impact on performance in the toy continuous environment (Figure 3a), i.e., that it can be neither too small nor too large. We decided to tune the *time unit* also for complex environments (Figures 3b, 8 and 9). The insight from the toy environment transferred to the complex case and there were gains of even $100\%$ in some cases over the default value of the time units used in the "expert-tuned" environments. A further insight to be had is that for simpler environments like the toy, *Pusher* and *Reacher*, the effect of the selection of the *time unit* was not as pronounced as for a more complex environment like *HalfCheetah*. This makes intuitive sense as one can expect a narrower range of values to work for more complex environments. This shows that it is even more important to tune such dimensions for more complex environments.

The *basic* agent design we showed above does this once and sets its optimal *time unit* statically. An ideal adaptive agent design would even set the *time unit* in an *online* manner. Since the trends from the toy environment coarsely transfer to the complex environments, coarse and quick insights can be gained on the toy environments.

**Varying *action range*** We observed similar trends as for *time unit*, in that there was an optimal value of *action range*, i.e., that it can be neither too small nor too large. Figure 9 shows this for all considered agents on HalfCheetah (for SAC and DDPG, runs for *action range* values $>= 2$ and $>= 4$ crashed and are absent from the plot). This supports the insight gained on our simpler environment that tuning this value may lead to significant gains for an agent. For already tuned environments, such as the ones in *Gym*, this dimension is easily overlooked but when faced with new environments setting it appropriately can lead to substantial gains. In fact, even in the tuned environment setting of *Gym*, we found that all three algorithms performed best for an *action range* $0.25$ times the value found in *Gym* for *Reacher* (Figures 8c, 8k, 8g in Appendix H). Moreover, the learning curves in Appendix N further show that for increasing *action range* the training gets more variant. The difference in performances across the different values of *action range* is much greater in the complex environments. We believe this is due to correlations within the multiple degrees of freedom as opposed to a rigid object in the toy environment.

To the best of our knowledge, the impact of *time unit* and *action range* is under-researched while developing agents because the standard environments have been pre-configured by experts. However, it's clear from Figure 3b, that pre-configured values were not optimal and even basic tuning improves performance significantly in even *known* environments. In a completely *unknown* environment, if we want agents to perform optimally, these dimensions would need to be taken into account even more when designing agents.

**Varying *transition noise*** We observe similar trends for injecting transition noise into Atari environments for all three agents as for the toy environments. We also observe that for some of the environments, transition noise actually helps improve performance. This has also been observed in prior work [61]. This happens when the exploration policy was not tuned optimally since inserting transition noise is almost equivalent to $\epsilon$-greedy exploration for low values of noise. We also observed a similar effect for the toy environments in Figure 18 in Appendix J. However, we also observe that performance drop is different for different environments. This is to be expected as there are other dimensions of hardness which we cannot control or measure for these environments.

**Varying *reward delay*** We see that on average performance drops for the delay experiments when more delay is inserted , as was the case for the toy environments. For *qbert* (Figure 3d), these drops are greater on average across the agents. However, for *breakout* (Figure 6b), in many instances, we don't even see performance drops. In *beam_rider* (Figure 6a) and *space_invaders* (Figure 6d), the magnitude of these effects are intermediate to *breakout* and *qbert*. This trend becomes clearer when we also look at Figures 7b-p in Appendix H. We believe this is because large delays from played action to reward are already present in *breakout*, which means that inserting more delays does not have as large an effect as in *qbert* (Figures 3d). Agents are strongest affected in qbert which, upon looking at gameplay, we believe has the least delays from rewarding action to reward compared to the other games. The trends for delay were noisier than for transition noise, even though on average the trends transferred from MDP Playground to the complex environments. Many considered environments tend to also have repetitive sequences which would dilute the effect of injecting delays. Many of the learning curves in Appendix N, with delays inserted, are indistinguishable from normal learning curves. We believe that, in addition to the motivating examples, this is empirical evidence that delays are already present in these environments and so inserting them does not cause the curves to look vastly different. In contrast, when we see learning curves for transition noise, we observe that, as we inject more and more noise, training tends to a smoother curve as the agent tends towards becoming a completely random agent.

Additionally, we also have experiments with similar trends also for another dimension - *reward noise*. The average rank correlation over 12 experiments (3 agents x 4 Atari environments) was $0.867$ for *transition noise*, $0.617$ for *reward delay*, and $0.733$ for *reward noise*. Tables 1, 2 and 3 list the individual rank correlation for each experiment, i.e. agent, environment and dimension.

To analyse transfer of dimensions between toy and complex benchmarks, for the Atari experiments, we use the Spearman rank correlation coefficient between corresponding toy and complex experiments for performance across different values of the dimension of hardness. The Spearman correlation was $>= 0.7$ for $19$ out of $24$ experiments and a positive correlation for four of the remaining five. DQN with delays added on breakout was the only experiment with correlation $0$.

**Varying *irrelevant features*** We observed that introducing *irrelevant dimensions* to the control problem, while keeping the number of relevant dimensions fixed to 2, decreased an agent's performance (see Figures 3c & 17f). This gives us the insight that having irrelevant features interferes with the learning process. An inductive bias that learns to focus only on the relevant dimensions could be unit-tested to gain coarse insights on the toy environments.

We have shown similar trends for SAC on HalfCheetah in Figure 9a in Appendix H.

**Varying Multiple Dimensions** In *MDP Playground*, it is possible to vary multiple dimensions at the same time in the same base environment. For instance, Figure 4d shows the interaction effect (an inversely proportional relationship) between the *action range* and the *time unit* in the continuous toy environment with DDPG. This insight allows us to design an adaptive agent which sets its *action range* depending on the *time unit* and vice versa. Since many real-world systems can be described in terms of a simple rigid body moving towards a target point, the toy continuous environment is a useful testbed for this.

More such experiments can be found in Appendix L, including varying both $P$ and $R$ *noise*s together in discrete environments and more. Further design ideas for new agents can be found in Appendix E.

## 4.2 Insights into Existing Agents

Apart from the insights gained for designing agents above, we discuss more insights for existing agents explicitly here.

The experiment for varying representations on toy environments discussed above (Figures 2a-c) further showed that the degradation in performance is much stronger for DQN compared to Rainbow and A3C which are known to perform better than DQN in complex environments.

This led us to another interesting insight regarding the inductive bias of CNNs. It was unexpected for us that the most problematic transform for the agents to deal with was *shift*. Despite the spatial invariance learned in CNNs [30], our results imply that that seems to be the hardest one to adapt to. As these trends were strongest in DQN, we evaluated further ranges for the individual transforms for DQN. Here, *shift*s had the most possible different combinations that could be applied to the images. Therefore, we quantised the *shift*s to have fewer possible values. Figure 2d shows that DQN's performance improved with increasing quantisation (i.e., fewer possible values) of *shift*. We noticed similar trends for the other transforms as well, although not as strong as they do not have as many different values as *shift* (see Figures 29b-c in Appendix J). We emphasize that in a more complex setting, we would have easily attributed some of these results to luck but in the setting where we had individual control over the dimensions, our platform allowed us to dig deeper in a controlled manner.

Another insight we gain is from the *time unit* experiment (see Figures 3a and 3b), which indicates *time unit* should not be infinitesimally small to achieve too fine-grained control since there is an optimal *time unit* for which we should repeat the same action [7].

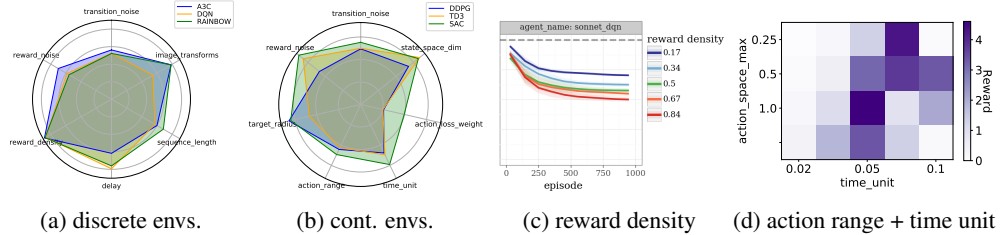

(a) discrete envs.        (b) cont. envs.        (c) reward density        (d) action range + time unit

Figure 4: Analysing and Debugging

In Figure 3d, where we varied *delay* on *qbert*, we show how a dimension induces hardness in an environment. This result is representative of the experiments on toy and complex environments which are included in Appendix H and H with the difference that results are noisier in complex environments since the dimensions are already present there in varying degrees. We, thus, studied what kinds of failure modes can occur when an agent is faced with such dimensions and even obtained noisy learning curves typically associated with RL on the *toy* environments as can be seen in Appendix M.

At the same time, the experiment in Figure 3d also shows how the complex environment wrappers allow researchers, who are curious, to study the robustness of their agents to these dimensions on complex environments, without having to fiddle with lower-level code. This is a typical use-case further down the agent development pipeline, i.e., close to deployment.

**Design and Analyse Experiments** We allow the user the power to inject dimensions into toy or complex environments in a fine-grained manner. This can be used to define custom experiments with the dimensions. The results can be analysed in an accompanying *Jupyter notebook* using the 1D plots. There are also radar plots inspired by bsuite [42], but with more flexibility in choosing the dimensions, and these can even be applied to complex environment experiments. Since, different users might be interested in different dimensions, these are loaded dynamically from the data. For instance, radar plots for the dimensions we varied in our toy experiments can be seen as in Figures 4a and 4b.

### 4.3 Debugging Agents

Analysing how an agent performs under the effect of various dimensions can reveal unexpected aspects of an agent. For instance, when using bsuite agents, we noticed that when we varied our environment's *reward density*, the performance of the bsuite Sonnet DQN agent would go up in proportion to the density (see Figure 4c). This did not occur for other bsuite agents. This seemed to suggest something different for the DQN agent and when we looked at DQN's hyperparameters we realised that it had a fixed $\epsilon$ schedule while the other agents had decaying schedules. Such insights

can easily go unnoticed if the environments used are too complex. The high bias nature of our toy environments helps debug such cases.

In another example, in one of the Ray versions we used, we observed that DQN was performing well on the *varying representations* environment while Rainbow was performing poorly. We were quickly able to ablate additional Rainbow hyperparameters on the toy environments and found that their noisy nets [13] implementation was broken (see Figure 5 in Appendix). We then tested and observed the same on more complex environments. This shows how easily and quickly agents can be debugged to see if something major is broken. This, in combination with their low computational cost, also makes a case to use the toy environments in Continuous Integration (CI) tests on repositories.

Further, we believe the same structured nature of *MDP Playground* also makes it a valuable tool for theoretical research. We evaluated tabular baselines Q-learning [52], Double Q-learning [60] and SARSA [52] on the discrete non-image based environments with similar qualitative results to those for deep agents. These can be found in Appendix K. This makes our platform a bridge between theory and practice where both kinds of agents can be tested.

The experiments here are only a glimpse into the power and flexibility of MDP Playground. Users can even upload custom $P$s and $R$s and custom images for representations $O$ and our platform takes care of injecting the other dimensions for them (wherever possible). This allows users to control different dimensions in the same base environment and gain further insights.

# 5   Discussion and Related Work

The *Behaviour Suite for RL* [bsuite; 42] is the closest related work to MDP Playground. [42] collect known (toy) environments from the literature and use these to characterise agents based on their performance on these environments. Most environments in *bsuite* can be seen as an intermediate step between our MDPs and more complex environments. This is because *bsuite*'s environments are already more specific and complex than the toy environments in *MDP Playground*. This makes *bsuite*'s dimensions not orthogonal and *atomic* like ours and thus not individually controllable. Fine-grained control is a feature that sets our platform apart. *bsuite* has a collection of *presets* chosen by experts which work well but would be much harder to play around with. While *MDP Playground* also has good presets through default values defined for experiments, it is much easier to configure. Further, it also means that *bsuite* experiments are much more expensive than ours. While *bsuite* itself is quite cheap to run, *MDP Playground* experiments are an order of magnitude cheaper. In contrast to *bsuite*, we demonstrate how the identified trends on the toy and complex environments can be used to design and debug agents. Further, *bsuite* currently has no toy environment for Hierarchical RL (HRL) agents while *MDP Playground*'s rewardable sequences fits very well with HRL. Finally, bsuite offers **no continuous control environments**, whereas MDP Playground provides both discrete and continuous environments. This is important because several agents like DDPG, TD3, SAC are designed for continuous control. A more detailed comparison with bsuite and other related work can be found in Appendix D.

Toybox [58] and Minatar [62] are also cheap platforms like ours with similar goals of gaining deeper insights into RL agents. However, their games target the specific *Atari* domain and are, like *bsuite*, more specific and complementary to our approach.

We found [3] the most similar work to ours in spirit. They propose that current deep RL research has been increasing the complexity of the dynamics $P$ but has not paid much attention to the state distributions and reward distribution over which RL policies work and that this has made RL agents brittle. This also raises concerns about the narrow scope of these so-called "complex" environments and we aim to remedy that with our dimensions. We agree with them in this regard. However, they only target continuous environments. We capture their dimensions in a different manner and offer many more dimensions with fine-grained control. Furthermore, their code is not open-source.

Further research includes *Procgen* [11], *Obstacle Tower* [24] and *Atari* [6]. Procgen adds various heterogeneous environments and tries to quantify generalisation in RL. In a similar vein, Obstacle Tower provides a generalization challenge for problems in vision, control, and planning. These benchmarks do not capture orthogonal dimensions of difficulty and as a result, they do not have the same type of fine-grained control over their environments' difficulty and neither can each dimension be controlled independently. We view this as a crucial aspect when testing new agents. [12] provides

some overlapping dimensions with our platform but it consists of only continuous environments, and doesn't target the toy domain.

# 6   Limitations of the Approach and its Ethical and Societal Implications

The toy environments are meant to be design and debug testbeds and not for engineering/tuning the final agent HPs. As such, they are extremely cheap compared to complex environments and (as one would expect), they can only be used to draw high-level insights that transfer and are likely not as discriminating as complex environments for many of the finer changes between RL agents. They also cannot be used directly to determine the values of hyperparameters (HPs) to use on complex environments. For example, just as complex environments require bigger NNs, they would need correspondingly different HPs, such as bigger replay buffers. Even the performance of agents in bsuite (which has more complex environments than our benchmark) do not transfer to the more complex environments (https://github.com/deepmind/bsuite/issues/14). In a similar vein, to the best of our knowledge, MNIST hyperparameters do not transfer to ImageNet and it is only used for testing out initial design ideas.

Further, high-dimensional control problems where there are interaction effects between degrees of freedom are not captured in the toy rigid body control problem as this is the domain of complex benchmarks and beyond the scope of this platform. (The platform does provide complex environment wrappers, though, which inject some of the mentioned dimensions. We couldn't find such wrappers in the literature/on the Internet.)

Finally, Multi-Agent RL, Multi Objective RL, Time Varying MDPs (and probably some more research areas) are beyond the scope of the current work.

In terms of the broader impact on society and ethical considerations, we foresee no direct impact, only indirect consequences through RL since our work promotes standardisation and reproducibility which should accelerate RL research. An additional environmental impact would be that, at least, prototyping and testing of agents could be done cheaply, reducing carbon emissions.

# 7   Conclusion and Future Work

We introduced a low-cost platform to design and debug RL agents and provided instructions on how to use it with supporting experiments. The platform allows us to disentangle various factors that make RL environments hard by providing fine-grained control over various dimensions. This also lends itself to easily achievable insights and helps debug agents. We further demonstrated how the performance of the studied agents is adversely affected by the dimensions. To the best of our knowledge, we are the first to perform a principled study of how significant aspects such as non-Markov information states, irrelevant features, representations and low-level dimensions, like time discretisation, affect agent performance.

We want *MDP Playground* to be a community-driven effort and it is open-source for the benefit of the RL community at https://github.com/automl/mdp-playground. While we tried to exhaustively identify dimensions of hardness, it is unlikely that we have captured *all* orthogonal dimensions in RL. We welcome more dimensions that readers think will help us encapsulate further challenges in RL and will add them based on the community's thoughts.

Future work can tackle not only theoretical development of such dimensions but also additional analysis of such dimensions in complex domains such as *Mujoco* and dexterous manipulation [46].

Given the current brittleness of RL agents [18], and many claims that have been challenged [5, 58], we believe RL agents need to be tested on a lower and more basic level to gain insights into their inner workings. *MDP Playground* is like a programming language for regularly structured MDPs which allows delving deeper into the inner workings of RL agents.

## Acknowledgements

The authors gratefully acknowledge support by BMBF grant DeToL, by the Bosch Center for Artificial Intelligence, and by the European Research Council (ERC) under the European Union's Horizon 2020 research and innovation programme under grant no. 716721, by the state of Baden-Württemberg through bwHPC and the German Research Foundation (DFG) through grant no INST 39/963-1 FUGG. They would like to thank their group, especially Joerg, Steven, Samuel, for helpful feedback and discussions. Raghu would like to additionally thank Michael Littman for his feedback and encouragement and the RLSS 2019, Lille organisers and participants who he had interesting discussions with.

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
