# OpenReview forum: "MDP Playground: A Design and Debug Testbed for Reinforcement Learning"
_NeurIPS.cc/2021/Track/Datasets_and_Benchmarks/Round1 — Submitted to NeurIPS 2021 Datasets and Benchmarks Track (Round 1)_

### Official Review · Reviewer_fvhY · 2021-07-04
**A comprehensive work providing tools for scrutinize Reinforcement Learning**

**Rating:** 7
**Confidence:** 5

**Strengths:**

- An exhaustive study
- Clearly outlines and motivates each dimensions along with RL agents should be investigates
- Detailed set of experiments in the Appendix
- Includes both continuous and discrete domains
- Clearly positions itself wrt to prior works
- I appreciate authors being clear about their computational and carbon footprint

**Weaknesses:**

- Claims at certain places are over-read and over-generalized -- for example its quite unlikely that the intuitions derived in Section4 using the toy envs will generalize more broadly. Intuitions derived here can be used as guiding principles (and I believe this is what authors wanted to convey). I'd recommend authors to revise the language and presentation to avoid mis-communication
- There is a growing concern in the RL community that we have overgrown past environments in OpenAIGym and dmControl. They are too simple and narrowly defined. Authors here however refer to 'HalfCheetah' as complex environment. Some recalibration of expectation is needed. This is especially critical since this work is posed as a "benchmark", if successful, will be used study and facilitate future developments. An uncalibrated benchmark has the potential to steer the field in the wrong direction.

**Additional Feedback:**

- Extending the analysis to a few complex domains [1] can be really valuable.

[1] @INPROCEEDINGS{Rajeswaran-RSS-18,
    AUTHOR    = {Aravind Rajeswaran AND Vikash Kumar AND Abhishek Gupta AND
                 Giulia Vezzani AND John Schulman AND Emanuel Todorov AND Sergey Levine},
    TITLE     = "{Learning Complex Dexterous Manipulation with Deep Reinforcement Learning and Demonstrations}",
    BOOKTITLE = {Proceedings of Robotics: Science and Systems (RSS)},
    YEAR      = {2018},
}
[2]

**Clarity:**

- easy to read and follow
- Appendix is too exhaustive with

**Correctness:**

- Paper is easy to read, well motivated in its design choices and thorough in its experiments.

**Documentation:**

- Please provide more details on the wrapper for the complex environments

**Ethics:**

- no direct ethical concerns
- no dataset is involved


**Relation To Prior Work:**

- Section 5 clearly positions itself wrt prior work in terms of scope and problem space

**Summary And Contributions:**

- Presents MDP playground -- a tool-kit/ platform  to study various design details of RL agents
- Outlines dimensions along with the RL agents can be investigated. Provides clean and easy to instantiate abstractions.
- Provides exhaustive studies investigating these dimensions, drawing insights and highlighting take-aways.

---

> ### Author Response · Authors · 2021-07-09
> **Response to Reviewer fvhY**
>
> > An exhaustive study … Clearly outlines and motivates each dimensions … Detailed set of experiments … both continuous and discrete domains … Clearly positions itself wrt to prior works ... clear about their computational and carbon footprint”
>
> Thank you very much for your kind words and strong support of our paper!
>
> >Claims at certain places are over-read and over-generalized -- for example its quite unlikely that the intuitions derived in Section4 using the toy envs will generalize more broadly. Intuitions derived here can be used as guiding principles (and I believe this is what authors wanted to convey). I'd recommend authors to revise the language and presentation to avoid mis-communication
>
> Apologies if our claims seemed too strong. We have updated the text to make the claims sound weaker and avoid mis-communication.
>
> >There is a growing concern in the RL community that we have overgrown past environments in OpenAIGym and dmControl. They are too simple and narrowly defined. Authors here however refer to 'HalfCheetah' as complex environment. Some recalibration of expectation is needed. This is especially critical since this work is posed as a "benchmark", if successful, will be used study and facilitate future developments. An uncalibrated benchmark has the potential to steer the field in the wrong direction.
>
> We agree with your statement. We didn’t pay enough attention to the term “complex” before. Would the reviewer think that the term “high-variance” environments is better? Would you have any additional suggestions on recalibration? We have added a sentence saying that the so-called complex environments actually are too narrow in some respects, as was also noted by [1].
>
> >Please provide more details on the wrapper for the complex environments
>
> Thank you for pointing this out. We have updated the docs here:
>
> [Gym](https://github.com/automl/mdp-playground/blob/master/mdp_playground/envs/gym_env_wrapper.py)
>
> [Mujoco](https://github.com/automl/mdp-playground/blob/master/mdp_playground/envs/mujoco_env_wrapper.py)
>
> > Extending the analysis to a few complex domains [1] can be really valuable.
>
> Thank you very much for the interesting research direction. We have added it to the future work section and look forward to more such interesting research being performed in the community.
>
> [1] Olov Andersson and Patrick Doherty. Toward robust deep rl via better benchmarks: Identifying neglected problem dimensions. In 2nd Reproducibility in Machine Learning Workshop at ICML 2018, Stockholm, Sweden, 2018.

---

### Official Review · Reviewer_qg1o · 2021-07-04
**A good testbed for RL but more justification is needed**

**Rating:** 6
**Confidence:** 4
**Correctness:** See the weakness seciton.

**Strengths:**

The paper proposes a rich set of variations of the RL tasks that can significantly affect the performance of an RL system but are often overlooked since they have been tuned by experts in standard RL benchmarks. This will bring more attention to research in this area to make RL systems more robust to such hyperparameters. Additionally, I find the explanation of the design decisions in Appendix H inspiring.

**Weaknesses:**

My major concern is, to what degree can the insights from the simple environments transfer to the more complex environments? The examples in section 4 are not convincing to me. For the image transformation example, what is the message of Figure 1a-c? Is the same observation that image transformation has a different effect on DQN and Rainbow observed in more complex environments? The example of varying time units seems obvious as a larger time unit means less control over the agent which will eventually hurt performance. An important question is how are the default values set for different dimensions of the tasks? For the high-level insights drawn from the experiments, are they consistent across other dimensions? For example, when evaluating the effect of the time unit, how much does the value of the diameter of the MDP affect the conclusion? If the high-level insights are consistent across different sets of default values, it becomes more convincing that they will also transfer to more complex environments.

**Additional Feedback:**

No additional feedback.

**Clarity:**

Discussion of the related work is clearly explained. A few parts of the experiments could be explained more clearly: Apart from the figure 1a-c mentioned above, it is also not clear to me what is the message of figure 2a-b.

**Documentation:**

There are sufficient details on the data collection and intended usages. However, I did not find a separate webpage that hosts the documentation.

**Relation To Prior Work:**

Yes.

**Summary And Contributions:**

The paper proposes a set of tasks for reinforcement learning with minimal complexity for quickly designing and debugging new algorithms. Different variations of the tasks are introduced to disentangle factors that affect an agent's learning performance. The paper shows a few examples of the transfer of the high-level insights drawn from the proposed tasks to the more complex tasks.

---

> ### Author Response · Authors · 2021-07-09
> **Response to Reviewer qg1o part 1**
>
> >The paper proposes a rich set of variations … make RL systems more robust ... Appendix H inspiring.
>
> Thank you very much for the kind words!
>
> >For the image transformation example, what is the message of Figure 1a-c? Is the same observation that image transformation has a different effect on DQN and Rainbow observed in more complex environments?
>
> Apologies that the text was unclear. The main takeaways are:
> 1) Regarding comparison between the agents, our experiment shows that Rainbow and A3C are more robust to the transforms as may be expected of them because they perform better than DQN (in several papers including [1], [2]) on complex environments. (We would like to point out that it is non-trivial to obtain strong conclusions about image transforms for complex environments, since the underlying task is not as simple as classifying the state for an object seen in the image (which is the case with our toy environments) and so we only make the weaker claim mentioned here.)
> 2) Regarding the image representations aspect of environments, it helps us discriminate between the usefulness of fully connected networks and CNNs for image observations without having to run on complex environments. Of course, this is well-known in the literature and is only mentioned in the paper to remind readers that our benchmark captures this aspect as well.
>
>
> >The example of varying time units seems obvious as a larger time unit means less control over the agent which will eventually hurt performance.
>
> We would only partly agree here. As shown in the experiments, a *smaller* time unit also hurts control performance. This is likely because the search space over the actions also grows much larger for smaller timesteps. For instance, say, we had an action space size of 10 and a time unit of, say, 100ms. Assume that optimal actions last for 100ms each (only to simplify the example). If we now reduce the time unit to 1ms then we have to search over 10^100 action sequences to arrive at an optimal action (sequence) whereas with the original time unit we would have had to search only over 10 actions to find the optimal action. (Of course, the situation is more complex in reality and different actions probably need repeating for different amounts of time, but for fixed discretizations of the time unit, we showed that the trend is similar for toy and complex environments, i.e., that neither too large nor too small a time unit is optimal.)
>
> Further, we have added new transfer experiments to the main paper (in red as mentioned in the common response to the reviewers) and would encourage you to take a look at those as well. We hope that the new experiments and insights allay your concerns.
>
> We would also further point out that it is good that aspects such as time unit and action range, which one might expect to have expected outcomes, are captured in our environments and make it easy for researchers to tackle these aspects on a smaller scale than having to use complex environments for these. Just to emphasize our point a bit more, we found several settings of time unit and action range to be sub-optimal in “expert-tuned” complex environments (please see the description of the new experiments in the main paper lines 216 and 231).
>
> >Apart from the figure 1a-c mentioned above, it is also not clear to me what is the message of figure 2a-b.
>
> Apologies for not being clear enough. The takeaway message for Figures 2a-b, is what we described regarding the time unit above - that the trend is similar for toy and complex environments, i.e., that neither too large nor too small a time unit is optimal. We have made the text clearer regarding this insight (line 214 In the new upload).
>
> >An important question is how are the default values set for different dimensions of the tasks?
>
> Thank you for pointing this out and apologies if this was unclear. We list here the defaults for the dimensions from the main paper (and we have also updated the [documentation](https://github.com/automl/mdp-playground/blob/master/mdp_playground/envs/rl_toy_env.py) to explicitly point out the default values for all dimensions). Where not mentioned otherwise, the dimensions are set to these values:
>
> Discrete environments:
> * Delay = 0
> * Sequence Length = 1
> * Transition noise = 0
> * Reward Noise = 0
> * Reward Density = 0.25
> * Diameter = 1
> * Irrelevant Features = None
> * Representations = Categorical, i.e., non-image
>
> Continuous environments:
> * Delay = 0
> * Sequence Length = N/A since it’s variable
> * Reward Density = N/A since it’s a fixed reward function, i.e., distance moved towards the goal in the last step
> * Transition noise = 0
> * Reward Noise = 0
> * Diameter = 2 * sqrt(2) * state_space_max (and state_space_max = infinity by default)
> * Irrelevant Features = None
> * Representations = Feature-space, i.e., non-image
> * Action Range = [-infinity, infinity]
> * Time Unit = 1.0
> * Target Radius = 0.05

---

> > ### Comment · Reviewer_qg1o · 2021-07-10
> > **Thanks**
> >
> > I appreciate the clarification from the authors. However, I agree with reviewer kz3Y that the current evidence does not convince me that improvement on playground can lead to improvement on more complex environments. As such, my score remains the same.

---

> > > ### Author Response · Authors · 2021-07-11
> > > **Response to Reviewer qg1o**
> > >
> > > Thanks so much for the prompt response! As mentioned in the response to Reviewer kz3Y, we are working on some experiments that we hope allay at least some of the concerns.

---

> ### Author Response · Authors · 2021-07-09
> **Response to Reviewer qg1o part 2**
>
> >For the high-level insights drawn from the experiments, are they consistent across other dimensions? For example, when evaluating the effect of the time unit, how much does the value of the diameter of the MDP affect the conclusion? If the high-level insights are consistent across different sets of default values, it becomes more convincing that they will also transfer to more complex environments.
>
> Thank you for pointing this out. Yes, such insights are consistent in the experiments we have seen so far. For instance, please see the experiments with 2 dimensions in Figure 4d (in the new upload). The action range always has an optimum which is neither too large nor too small for different values of the time unit. In fact, this figure gives an even deeper insight - the optimal action range and the time unit are anti-correlated, i.e., one is allowed to have a greater range of actions for smaller time units. This makes sense because for larger time units if one performs too large an action, the search for optimal actions is likely to diverge more easily.
>
> Similar insights regarding consistent insights across different values of dimensions can also be seen in Figures 35-37 and 42 in the appendix. In Figures 43a and b, we have even shown this across 3 dimensions: correlations between 2 dimensions (action range and time unit) are similar across a 3rd dimension - the order of the transition dynamics.
>
>
>
> >However, I did not find a separate webpage that hosts the documentation.
>
> Unfortunately, it seems the link for the documentation (https://automl.github.io/mdp-playground/) was not so prominent on the Github page. It was under the “About” section on the top right of the webpage. We have now also added it in a more prominent location in the README, including links to the most important classes’ documentations. We hope this allays the concern.
>
> [1] M. Hessel, J. Modayil, H. van Hasselt, T. Schaul, G. Ostrovski, W. Dabney, D. Horgan, B. Piot, M. Azar, and D. Silver. Rainbow: Combining improvements in deep reinforcement learning. In S. A. McIlraith and K. Q. Weinberger, editors, Proceedings of the Conference on Artificial Intelligence (AAAI’18), pages 3215–3222. AAAI Press, 2018.
>
> [2] V. Mnih, A. P. Badia, M. Mirza, A. Graves, T. P. Lillicrap, T. Harley, D. Silver, and K. Kavukcuoglu. Asynchronous methods for deep reinforcement learning. In M. Balcan and K. Weinberger, editors, Proceedings of the 33rd International Conference on Machine Learning (ICML’16), volume 48, pages 1928–1937, 2016.

---

### Official Review · Reviewer_kz3Y · 2021-07-04
**Interesting benchmark for an important niche, but not clear that insights generalize to complex environments**

**Rating:** 5
**Confidence:** 4

**Strengths:**

The biggest strength of the MDP playground is that it enables quick iteration and feedback when designing RL algorithms: this is a niche that to my knowledge had not been filled before (although arguably bsuite may count, though it does still take longer to run than the MDP playground). The framework appears to be very flexible, while still being easy to use, and covers a wide range of RL use cases, which should give it broad appeal.

**Weaknesses:**

In my view the paper suffers from two major limitations that drive my low score. First, the paper is quite dense and unclear. I expand on this in the section on clarity, including some recommendations for better exposition which would allow the reader to better grasp the ideas in the paper.

Second, the utility of the MDP playground depends on the key claim that algorithm performance on the MDP playground is predictive of performance in more complex environments. However, there does not seem to be much evidence presented for this key claim: the evidence seems to be a single experiment which shows that the time unit is an important parameter in the MDP playground as well as in the HalfCheetah. More evidence needs to be presented to establish this key claim. Currently, I would not feel enthusiastic about using the MDP playground myself, because I don’t feel confident that the performance of an algorithm on the playground has much bearing on the performance of the algorithm in the setting I actually care about.

(In contrast, the other experiments presented on irrelevant features and multiple dimensions don’t seem very important to me, since the authors do not test whether the proposed “insight” generalizes to more complex environments.)

A more compelling experiment could look as follows:
1. Choose some complex environment (such as the HalfCheetah).
2. Determine which of the dimensions in the MDP playground could apply to that environment (there should be at least 3, and preferably 5+, for the experiment to be compelling).
3. Choose 2-3 RL algorithms.
4. Run the RL algorithm on the MDP playground to find how performance varies across the various dimensions. Do the same for the more complex environment (note this means you’ll have to modify the complex environment to allow the dimension to be varied). Check whether the results in the MDP playground are predictive of those in the more complex environment.

**Additional Feedback:**

N/A

**Clarity:**

I found the paper dense and hard to understand; the authors have crammed in a lot of content that is explained very briefly without clear examples and figures. For example, it took me until halfway through the paper to understand that these are what we might call “graphical” (PO)MDPs, rather than (say) gridworlds.

Some recommendations to improve clarity:
1. Choose half of the content from the paper, explain that half very well, and move the remaining half into appendices (or cut it entirely). See subsequent sections for how to do so.
2. Rather than explaining all ten dimensions with a paragraph each, start with a list of the ten dimensions with just one sentence explaining what that dimension is. Then choose 3-5 dimensions to explain in more depth. Include any dimensions that you intend to reference in your experiments (e.g. the time unit dimension).
3. Include a figure of a graphical (PO)MDP that might be generated by the playground. For each dimension that you explain in depth, explain with reference to specific states, actions and transitions in the figure, what the dimension corresponds to and how it can be controlled in your framework. An example of this type of exposition can be found in [1].
4. Merge Sections 2 and 3, so that you only explain the details of each dimension once. Once I got to Section 3, I had already forgotten what had been said about that dimension in Section 2 (because I had read about ten other dimensions in between!)
5. This is less related to the other points, but please show learning curves for agents rather than just AUC numbers (for example in Figures 1 and 2). These should take roughly the same amount of space, as you can plot multiple learning curves on the same graph.

[1] Turner, Alexander Matt, et al. "Optimal Policies Tend to Seek Power." arXiv preprint arXiv:1912.01683 (2019).

**Correctness:**

To my knowledge, the claims made are correct, apart from some minor nits mentioned below. However, as mentioned in the weaknesses, I am not convinced that the MDP playground is useful, though this may be more an issue of clarity.

Nits:

> An implicit assumption for many agents is that rewards are immediate depending on only the current information state and action.

This does not seem correct to me. It is well known that one of the main challenges in RL is credit assignment, and algorithm design takes this challenge into account. The entire point of credit assignment is that rewards are not immediate.

> This was done exhaustively to allow as many dimensions as possible

To claim that this was done exhaustively you should have some argument that there are no other dimensions that you have failed to find. This seems very hard to argue (and probably not true).

It is of course true that the MDP playground has identified a very large number of dimensions, and you should emphasize that -- it is only the word “exhaustively” that I take issue with.

**Documentation:**

Yes, the paper is fine on this axis (particularly as they have open sourced their code).

**Ethics:**

No.

**Relation To Prior Work:**

Yes. Bsuite is the obvious benchmark to compare to, and the authors discuss it in detail. I might recommend that the authors emphasize insights that couldn’t have been gained from bsuite (for example, I believe the insight about the time unit being important could not have come from bsuite).

**Summary And Contributions:**

(EDIT: Score raised from 4 to 5 after author rebuttal)

The authors propose and develop the MDP playground: a codebase which can generate random graphical POMDPs. The properties of the POMDPs can be controlled using tens of knobs that have been specifically selected to highlight particular challenges of reinforcement learning. They then demonstrate how the MDP playground can be used to generate insights about the performance of RL algorithms that can then be transferred to more complex settings.

---

> ### Author Response · Authors · 2021-07-09
> **Response to Reviewer kz3Y part 1**
>
> >The biggest strength of the MDP playground … quick iteration … very flexible … easy to use ... give it broad appeal.
>
> Thank you very much for the kind words!
>
> >A more compelling experiment could look as follows:
> .
> .
> .
> Do the same for the more complex environment ... Check whether the results in the MDP playground are predictive of those in the more complex environment.
>
> Thank you very much for the suggestion! We believe we have exactly these experiments in Appendix F, which we have now moved to the main paper (highlighted as red text in the new upload’s section 4.1). To sum up the high-level insights from there (more low-level insights are present in the text):
> * We observed on both, the toy and complex environments, that performance can improve when injecting transition noise. This has also been observed in prior work on complex environments [1].
> * We observed similar trends on toy and complex environments also for varying the size of the action range. This insight is similar to the one for the time unit dimension in that there is an optimal size of the action range for which the performance of the agents reached a peak.
> * To analyse the transfer of dimensions between toy and complex benchmark experiments, we used the Spearman rank correlation coefficient between corresponding toy and complex experiments for performance across different values of the dimension of hardness. The Spearman correlation was >= 0.7 for 19 out of 24 experiments and a positive correlation for four of the remaining five.
>
>
> Additionally, we would like to point out that transfer from toy to complex environments is not the only interesting use-case for MDP Playground. As discussed in section 4.3:
> * Varying key dimensions as done in MDP Playground can be used to study agents in more depth as was the case with the reward sparsity experiment for the bsuite DQN agent (Figure 4c in the new upload).
> * Unit testing of agents with MDP Playground, as for the Rainbow agent in Ray, revealed that the noisy nets implementation was broken (Figure 5 in the new upload). We also observed this later on the complex environments.
>
>
> >An implicit assumption for many agents is that rewards are immediate depending on only the current information state and action.
>
> >This does not seem correct to me. It is well known that one of the main challenges in RL is credit assignment, and algorithm design takes this challenge into account. The entire point of credit assignment is that rewards are not immediate.
>
> Apologies for the poor wording. What we meant to say was that the design of many current agents doesn’t take into account that the agent state they use may be only partially observable. For example, DQN just stacks, say, the last 4 observations and uses it as the agent’s state, i.e., DQN applies Q-learning using an agent state that may be only partially observable. This goes against the definition of the state used in Q-learning which is assumed to be fully observable, so the value functions learnt may not be correct for the true underlying state. We called this an implicit assumption because determining the fully observable state is not (usually) explicitly part of the agent design. We have removed the sub-optimal wording from the text and now only describe the dimension.
>
>
> >This was done exhaustively to allow as many dimensions as possible
>
> >To claim that this was done exhaustively you should have some argument that there are no other dimensions that you have failed to find. This seems very hard to argue (and probably not true).
>
> Apologies again for the poor wording. You are right that we have most likely not identified all dimensions exhaustively and we have modified the text to say that we try to exhaustively identify the dimensions. Would you agree with this re-wording? (We would like to mention that already in the initial submitted version, on line number 399, we invited readers to submit more dimensions in line with the community-driven nature of MDP Playground that we envision.)
>
>
> >Some recommendations to improve clarity:
>
> Thank you very much for putting in the time to suggest all the changes to help with the paper’s clarity.

---

> > ### Comment · Reviewer_kz3Y · 2021-07-09
> > **Confused about the new evidence**
> >
> > To recap, my review score primarily depends on whether the following claim is true, which I see as central to the utility of this benchmark:
> >
> > _Key Claim:_ Performance of an algorithm on the MDP playground is predictive of its performance on more complex environments.
> >
> > (I'm happy to take MuJoCo and Atari as "complex environments" for this purpose.)
> >
> > Thanks for highlighting more evidence about this claim that I hadn't previously seen. However, it doesn't seem very compelling to me. As I understand it, the main pieces of evidence are:
> >
> > 1. If we use the image representations in the MDP playground, we can show that CNNs have a good inductive bias for images.
> > 2. Experiments with MDP playground suggested that the time unit is an important hyperparameter, and this is indeed true of complex environments.
> > 3. A similar story holds for the "action range" hyperparameter.
> > 4. Experiments with MDP playground suggest that increased transition noise helps, and this is also true of complex environments.
> > 5. Reward delay matters in MDP playground, but doesn't have much of an effect in Atari. The authors hypothesize that Atari already has a lot of reward delay, so adding additional reward delay doesn't change much.
> > 6. Irrelevant features decrease learning performance, and in principle we could test new algorithms for dealing with this problem on MDP playground.
> >
> > (Section 4.2 mentions the difficulty of "shift" on the MDP playground, but doesn't demonstrate it on Atari, so it doesn't seem relevant to the Key Claim. Indeed, I am not sure whether this result will generalize to Atari or not.)
> >
> > 1 and 6 do not seem very compelling to me. The key question is whether a solution that improves results on the MDP playground _will then also work on more complex environments_. This hasn't been established, as far as I can tell.
> >
> > 2 and 3 are somewhat compelling. That being said, "we found a hyperparameter that gives improvements if tuned" is not _that_ interesting: _most_ hyperparameters give improvements if tuned. As I mentioned in my initial review, I'd be much more compelled by an experiment that tested a _set_ of dimensions, and saw whether the most important dimensions according to the MDP playground were also the most important dimensions in MuJoCo / Atari.
> >
> > I don't understand 4. Looking at Figure 16, it seems that on the MDP playground, transition noise has very little effect at all, and when it does have an effect, it tends to _decrease_ performance (for example, in DQN, with reward noise 5, if you go from transition noise 0.02 to 0.10, performance decreases). This seems like evidence against the Key Claim, not in favor of it.
> >
> > 5 is not compelling, since the effect in MDP playground mostly wasn't present in Atari. The authors' explanation of this result sounds plausible, but it still seems like in this case a researcher would have not benefitted from the playground at best, and would have been led astray by the playground at worst.
> >
> > Overall I still remain unconvinced of the key claim.
> >
> > ----
> >
> > I do think I underestimated the value of the MDP playground for debugging agents (Section 4.3) in my initial review. For that reason, and because of the improved clarity, I'm raising my score from a 4 to a 5.

---

> > > ### Author Response · Authors · 2021-07-11
> > > **Request for some clarification from Reviewer kz3Y**
> > >
> > > Thanks so much for the prompt response and the detailed clarification of your point of view! Thank you also for increasing your score! We have started an experiment and will respond in more detail after starting some more. In the meantime, we would be very grateful if you could clarify a few things.
> > >
> > > >I'd be much more compelled by an experiment that tested a set of dimensions, and saw whether the most important dimensions according to the MDP playground were also the most important dimensions in MuJoCo / Atari.
> > >
> > > Did you mean that we inject the set of dimensions at the *same* time in a single environment (and do this for both toy and complex)? Could you please clarify how you would propose to judge the importance of a dimension in this case? By ablating it from the same environment which had all dimensions injected and comparing the performance drops?
> > >
> > > Or did you mean that we inject the set of dimensions *one* at a time in a single environment (and do this for both toy and complex)? Could you please clarify how you would propose we judge the importance of a dimension in this case? By taking the average performance drop of a dimension as a measure its importance?
> > >
> > > Both the options we just described for the experiment, we believe, are problematic. Because they depend a lot on how much of what dimension is injected. For instance, one could in principle, "tune" the values of the dimensions injected in the toy environments to match the observed importances on the complex environments.

---

> > > > ### Comment · Reviewer_kz3Y · 2021-07-11
> > > > **Good point**
> > > >
> > > > > Or did you mean that we inject the set of dimensions one at a time in a single environment (and do this for both toy and complex)?
> > > >
> > > > I meant this one.
> > > >
> > > > > For instance, one could in principle, "tune" the values of the dimensions injected in the toy environments to match the observed importances on the complex environments.
> > > >
> > > > This is a good point. Perhaps you could imagine something more like a case study:
> > > > 1. Choose a new complex environment that you haven't run experiments on before.
> > > > 2. Describe a hyperparameter tuning strategy that you would run on that environment.
> > > > 3. Run any experiments you want with MDP playground.
> > > > 4. Based on these experiments, refine the hyperparameter tuning strategy to reduce the amount of tuning by (say) a factor of k^d (where d is the number of dimensions, and k is some constant, e.g k = 2).
> > > > 5. Run the original (expensive) strategy, a version of the original strategy where every dimension is made k times coarser (so that the tuning takes k^d less time), and the tuning strategy from point 4. Ideally, you would find that the point 4 strategy performs almost as well as the expensive strategy and much better than the coarse strategy.
> > > > 6. Make sure that the steps were done in this order, without later steps influencing earlier ones. Emphasize in the paper.
> > > >
> > > > (The underlying principle I'm using is "try to test the utility of the framework as directly as possible".)

---

> > > > > ### Author Response · Authors · 2021-07-14
> > > > > **Clarification about misunderstandings regarding some of the results**
> > > > >
> > > > > Thank you very much for your latest clarification and for the suggestion of a new experiment! We would like to clarify some misunderstandings and respond to the previous comments by the reviewer. (We have uploaded the latest version of the paper with the changes highlighted in blue. New Tables 1, 2 and 3 listing the individual rank correlation are not highlighted in blue.)
> > > > >
> > > > > >Reward delay matters in MDP playground, but doesn't have much of an effect in Atari. The authors hypothesize that Atari already has a lot of reward delay, so adding additional reward delay doesn't change much.
> > > > >
> > > > > >5 is not compelling, since the effect in MDP playground mostly wasn't present in Atari. The authors' explanation of this result sounds plausible, but it still seems like in this case a researcher would have not benefitted from the playground at best, and would have been led astray by the playground at worst.
> > > > >
> > > > > There seems to have been some misunderstanding. Reward delays DO matter in both, MDP Playground *and* Atari. However, the effect of delays was not *as strong* on Atari. And it was weaker than the effect of transition noise on Atari. Our apologies that we didn't *explicitly* mention in the high-level takeaways in our previous comment that the trends for transition noise and delay *are* similar to those on MDP Playground. We have tried to stress this more in the current paper upload.
> > > > >
> > > > > For the experiments with transition noise, the average Spearman rank correlation was 0.867 over 12 experiments; while for the experiments with delay, the average Spearman rank correlation was *only* 0.617 over 12 experiments, which is still a pretty strong correlation. The only really weak rank correlation (out of 24 total experiments) was for the DQN on *breakout* experiment, where the rank correlation was 0. This is why we tried to provide plausible reasoning across the 4 tested Atari environments as to why the trends were weak on breakout while pretty strong in the other Atari environments, *not* because the trends were not similar to the toy environments.
> > > > >
> > > > > Additionally, we now have experiments with similar trends also for *another* dimension - reward noise, where the average rank correlation over 12 experiments was 0.733. This lends yet more support for insights that were gained on MDP Playground being relevant for complex environments. We have added Tables 1, 2 and 3 listing the individual rank correlation for each experiment, i.e., for each agent, environment and dimension.
> > > > >
> > > > >
> > > > > >I don't understand 4. Looking at Figure 16, it seems that on the MDP playground, transition noise has very little effect at all, and when it does have an effect, it tends to decrease performance (for example, in DQN, with reward noise 5, if you go from transition noise 0.02 to 0.10, performance decreases). This seems like evidence against the Key Claim, not in favor of it.
> > > > >
> > > > > We would like to stress that we said that noise *can* help evaluation performance, it does not always help. Having no performance loss (or *sometimes* even small increases) in many cases, not *all* (if you look at Figure 16 (Figure 18 in the latest upload)) was surprising for us. Even for transition noise as high as 0.25 (in the column with reward noise 5 for DQN), there’s barely any drop in evaluation performance. We attributed this effect to low exploration in the absence of noise, which was also the same reasoning as the authors in [1]. Even in [1], the performance can drop or rise when injecting noise.
> > > > >
> > > > > >1 and 6 do not seem very compelling to me. The key question is whether a solution that improves results on the MDP playground will then also work on more complex environments. This hasn't been established, as far as I can tell.
> > > > >
> > > > > What we wanted to compare for "1" was inductive biases such as CNNs and Fully Connected Networks (FCNs). You are right that this has not been shown in an experiment. This is because we thought that the difference between the CNNs and FCNs was well established in the literature and thus only showed CNN performance. Would adding an experiment comparing CNNs and FCNs on the toy and complex environments allay this concern?
> > > > >
> > > > > Regarding "6", we have begun the irrelevant features experiment also on SAC on HalfCheetah and initial results (~2.4M timesteps run so far = 80 % of the budget from the main paper experiments) suggest a clear trend towards worse performance with more irrelevant features (as was the case on the toy environments) (please see Figure 9a in the current upload). Would this make "6" *more* compelling for you?
> > > > >
> > > > > (We know this is not the demonstration of a solution. However, if one had the ground truth of relevant features, clearly one could focus on only the relevant features in the environment and then the performance would improve on both, the toy and complex environments (as can be seen by the experiments Figures 3c and 9a in the current version of the paper where there were no irrelevant features) and MDP Playground provides the ground truth to test this.)

---

> > > > > > ### Author Response · Authors · 2021-07-14
> > > > > > **Clarification about misunderstandings regarding the positioning of MDP Playground**
> > > > > >
> > > > > > >Key Claim: Performance of an algorithm on the MDP playground is predictive of its performance on more complex environments.
> > > > > >
> > > > > > >Run the original (expensive) strategy, a version of the original strategy where every dimension is made k times coarser (so that the tuning takes k^d less time), and the tuning strategy from point 4. Ideally, you would find that the point 4 strategy performs almost as well as the expensive strategy and much better than the coarse strategy.
> > > > > >
> > > > > > We think the (reviewers kz3Y + qg1o) and (authors + reviewer fvhY) differ in the interpretation of the strength of the key claim. We *don't* believe that MDP Playground currently can be used for tuning of hyperparameter values for *domain experts*. We do believe that that has a lot of potential and we have been working on it, but because of how cheap MDP Playground is, we believe that that involves a lot of future work. We have made the limitation clearer, than it was previously, in the current version of the paper.
> > > > > >
> > > > > > To substantiate why we think the key claim should be weaker and why MDP Playground is still useful, we would like to briefly mention two motivating points that initiated MDP Playground's development:
> > > > > >
> > > > > > **A lot of performance gains in RL are due to code level tricks and tuning/engineering the algorithm**:
> > > > > > * The gains in complex environments in several instances need special tricks. For example, please see here: https://github.com/deepmind/bsuite/issues/14, even the performance of agents in bsuite (which has more complex environments than our benchmark) does not transfer to the more complex environments. Likewise, to the best of our knowledge, MNIST hyperparameters do not transfer to ImageNet and it is only used for testing out initial design ideas.
> > > > > > * Many gains in performance actually come from code-level optimisations and not algorithmic improvements: [3], [4].
> > > > > > What we want to say is that, in our opinion, performing "well" on complex environments may always need engineering and the engineering tricks employed don't always generalise to other complex domains, as another submission to this track points out [7]. For this reason, we believe the community needs to analyse and dig deeper into RL agents with environments designed as mentioned in Appendix G: Design Decisions (it was Appendix H in the old paper), without the "noise" of the current complex RL environments. It's debatable whether even the complex environments are complex enough and may be leading researchers astray [8]. Given the literature we have pointed out and the brittleness of RL agents, we believe we currently need to analyse RL as deeply as possible from the ground up.
> > > > > >
> > > > > > **General Unit-testing**
> > > > > >
> > > > > > As detailed in the current Appendix G, current RL agents were designed for *general* environments and they deserve some form of *general* testing as well. We know of no other benchmark that does this form of general testing (even bsuite has more specific environments). The intent of MDP Playground is that users can design something and show it as a proof of concept that the new design works against a dimension/partial observability. This is just like Q-learning [5] and double Q-learning [6] which were shown as proofs of concept on tabular or smaller environments several years before the advent of deep RL and complex environments. If those proofs of concept hadn’t been shown, deep RL wouldn’t have had the chance to engineer a solution.
> > > > > >
> > > > > >
> > > > > > [1] T. Wang, X. Bao, I. Clavera, J. Hoang, Y. Wen, E. Langlois, S. Zhang, G. Zhang, P. Abbeel, and J. Ba. Benchmarking model-based reinforcement learning. arXiv:1907.02057 [cs.LG], 2019.
> > > > > >
> > > > > > [2] van Hasselt, H.; Guez, A.; and Silver, D. 2016. Deep reinforcement learning with double Q-learning. In Proc. of AAAI, 2094–2100.
> > > > > >
> > > > > > [3] Logan Engstrom et al. Implementation Matters in Deep RL: A Case Study on PPO and TRPO. In: International Conference on Learning Representations. 2019. https://openreview.net/forum?id=r1etN1rtPB
> > > > > >
> > > > > > [4] Marcin Andrychowicz et al. What Matters for On-Policy Deep Actor-Critic Methods? A Large-Scale Study. In: International Conference on Learning Representations. 2021. https://openreview.net/forum?id=nIAxjsniDzg
> > > > > >
> > > > > > [5] C. J. C. H. Watkins. Learning from delayed rewards. PhD thesis,
> > > > > > University of Cambridge England, 1989.
> > > > > >
> > > > > > [6] H. van Hasselt. Double Q-learning. Advances in Neural Information Processing Systems, 23:2613–2621, 2010.
> > > > > >
> > > > > > [7] Mostafa Dehghani, Yi Tay, Alexey A. Gritsenko, Zhe Zhao, Neil Houlsby, Fernando Diaz, Donald Metzler, Oriol Vinyals. The Benchmark Lottery (https://openreview.net/forum?id=5Str2l1vmr-)
> > > > > >
> > > > > > [8] Olov Andersson and Patrick Doherty. Toward robust deep rl via better benchmarks: Identifying neglected problem dimensions. In 2nd Reproducibility in Machine Learning Workshop at ICML 2018, Stockholm, Sweden, 2018.

---

> ### Author Response · Authors · 2021-07-09
> **Response to Reviewer kz3Y part 2**
>
> >Rather than explaining all ten dimensions with a paragraph each, start with a list of the ten dimensions with just one sentence explaining what that dimension is. Then choose 3-5 dimensions to explain in more depth. Include any dimensions that you intend to reference in your experiments (e.g. the time unit dimension)
>
> Thank you for the suggestion. We have made the suggested changes to an extent (we moved dimensions that did not have experiments in the main paper to the Appendix; this has meant there are 2 fewer dimensions in the new upload). Since the paper is so broad in its scope, reviewers tend to have varied opinions on how much of what to include (as can be seen by the high variance in the review scores) and many previous reviewers preferred more dimensions and their descriptions in the main paper. Additionally, we felt since this is a benchmark track, it’d be better to describe the benchmark and its dimensions more. We would kindly request to, therefore, be a bit more gentle in judging our paper on this criterion.
>
> >Include a figure of a graphical (PO)MDP that might be generated by the playground. For each dimension that you explain in depth, explain with reference to specific states, actions and transitions in the figure, what the dimension corresponds to and how it can be controlled in your framework. An example of this type of exposition can be found in [1].
>
> Thank you for the suggestion. We have added figures where possible as you suggested (Figure 1 in the new upload).
>
> >Merge Sections 2 and 3, so that you only explain the details of each dimension once. Once I got to Section 3, I had already forgotten what had been said about that dimension in Section 2 (because I had read about ten other dimensions in between!)
>
> Thank you for the suggestion. We have made the suggested changes.
>
> >This is less related to the other points, but please show learning curves for agents rather than just AUC numbers (for example in Figures 1 and 2). These should take roughly the same amount of space, as you can plot multiple learning curves on the same graph.
>
> Thank you for the suggestion. We will add this very soon as well.
>
> >The authors propose and develop the MDP playground: a codebase which can generate random graphical POMDPs
>
> We would like to mention that this is only true for the discrete environments and the randomness is only true for assigning next states to transitions (since the grid-like structure of the MDP is fixed given the values set for the dimensions). However, because the environments also have image representations (even though the underlying POMDPs can be described as graphical) and the continuous environments are not random, we have refrained from calling them random graphical POMDPs. We have added a line to the text to say that the underlying POMDPs can be described as graphical POMDPs.
>
> [1] T. Wang, X. Bao, I. Clavera, J. Hoang, Y. Wen, E. Langlois, S. Zhang, G. Zhang, P. Abbeel, and J. Ba. Benchmarking model-based reinforcement learning. arXiv:1907.02057 [cs.LG], 2019.

---

> > ### Comment · Reviewer_kz3Y · 2021-07-09
> > **Thanks**
> >
> > Thanks for these changes; the new paper seems much more readable to me.

---

### Author Response · Authors · 2021-07-09
**Common Response to Reviewers**

We would like to thank the reviewers for their helpful feedback and for their time and effort, and we hope that they and their loved ones are all safe and healthy.

We now address the individual concerns of the reviewers. Please note that we have uploaded an updated version addressing several of the concerns that were raised. The remaining concerns which take a bit longer to fix will soon be updated as well. The major concerns, however, we believe have been addressed and we would be grateful if the reviewers provide feedback on whether we have correctly addressed their concerns.

Most of the changes to the paper are highlighted as red text in the paper (the new Figure 1 and merging most of Section 3 into Section 2 are not highlighted in red).
We have also updated the documentation for the GitHub repo (https://github.com/automl/mdp-playground).

---

### Author Response · Authors · 2021-07-15
**Summary of changes and final thank you to all the reviewers**

We would like to heartily thank the reviewers for all their hard work and help in improving our paper, especially for the prompt and detailed responses and putting in thought to suggest experiments for us! We hope that they will decide favourably with regard to our paper.

Summary of changes:

* Performed additional experiments to show transfer of the reward noise dimension to 4 Atari environments with 3 RL agents. (Section 4.1 and parts of Figures 6 and 7 in the current upload)
* Performed additional experiment to show transfer of the irrelevant features dimension to a complex environment. (Section 4.1 and Figure 9a)
* Improved clarity and presentation of the paper with a new figure and tables. (Added Figure 1 and Tables 1, 2 and 3. Merged the old Section 3 into Section 2)
* Most changes to the paper are highlighted as red and blue text in the paper (the new Figure 1, new Tables 1, 2 and 3 and merging most of Section 3 into Section 2 are not highlighted).
* We have also improved and updated the documentation for the GitHub repo (https://github.com/automl/mdp-playground).

The main difference of views between (authors + reviewer fvhY) and (reviewers kz3Y + qg1o), as we see it, is on the transfer of insights to complex environments, where we believe stronger transfer of insights with the very cheap toy environments of MDP Playground (experiments take *seconds* on the toy environments vs *days* with the standard complex environments) involve more future work and it is an exciting research direction to increase the fidelity of such cheap environments with more expensive environments. We hope our latest arguments managed to convince reviewers kz3Y and qg1o about why MDP Playground is important.

We would stress that:
* There are still several interesting insights that did transfer from the toy environments in MDP Playground to the complex environments (even if hyperparameters likely do not transfer from the very cheap toy environments in MDP Playground to the much more expensive complex environments of Atari and Mujoco).
* The toy environments in MDP Playground provide *general* unit testing for RL agents which are usually designed for *general* environments. This helps us characterise RL agents along several dimensions which helps us analyse them better.
* MDP Playground provides debugging capabilities for RL agents, with the ground truth being available.
* MDP Playground has not just toy environments but also complex environment wrappers which help study robustness of RL agents to dimensions of hardness in complex environments. We have performed extensive experiments even with these and analysed them in the paper.

---

### Decision · Program_Chairs · 2021-07-26

**Decision:**

Reject

**Comment:**

In this paper, the authors propose a new framework for testing RL algorithms. In particular, the framework offers knobs for independently varying many aspects that make RL tasks difficult (e.g. density of rewards or stochasticity).  Reviewer feedback centered on (a) clarity and (b) significance.  The authors are of course strongly encouraged to continue to improve the clarity of their paper, and to make sure claims are appropriately precise and contextualized.  With respect to significance, a consistent concern was whether progress on the synthetic tasks would generalize to progress on more realistic tasks.  The authors are encouraged to either add more clear evidence that the tasks are good benchmarks of progress and/or to make clearer that the dominant purpose of the playground is for understanding RL algorithms (e.g. debugging and controlled experiments) rather than benchmarking.  I'd encourage the authors to revise and resubmit.